# Fuzzy Alignments in Directed Acyclic Graph for Non-Autoregressive Machine Translation

**Zhengrui Ma**[1,2]**, Chenze Shao**[1,2]**, Shangtong Gui**[1,2]**, Min Zhang**[3] **& Yang Feng**[1,2*]

[1] Key Laboratory of Intelligent Information Processing
   Institute of Computing Technology, Chinese Academy of Sciences
[2] University of Chinese Academy of Sciences
[3] Harbin Institute of Technology, Shenzhen

## Abstract

Non-autoregressive translation (NAT) reduces the decoding latency but suffers from performance degradation due to the multi-modality problem. Recently, the structure of directed acyclic graph has achieved great success in NAT, which tackles the multi-modality problem by introducing dependency between vertices. However, training it with negative log-likelihood loss implicitly requires a strict alignment between reference tokens and vertices, weakening its ability to handle multiple translation modalities. In this paper, we hold the view that all paths in the graph are fuzzily aligned with the reference sentence. We do not require the exact alignment but train the model to maximize a fuzzy alignment score between the graph and reference, which takes captured translations in all modalities into account. Extensive experiments on major WMT benchmarks show that our method substantially improves translation performance and increases prediction confidence, setting a new state of the art for NAT on the raw training data.[1]

## 1 Introduction

Non-autoregressive translation (NAT) (Gu et al., 2018) reduces the decoding latency by generating all target tokens in parallel. Compared with the autoregressive counterpart (Vaswani et al., 2017), NAT often suffers from performance degradation due to the severe *multi-modality problem* (Gu et al., 2018), which refers to the fact that one source sentence may have multiple translations in the target language. NAT models are usually trained with the cross-entropy loss, which strictly aligns model prediction with target tokens. The strict alignment does not allow multi-modality such as position shifts and word reorderings, so proper translations are likely to be wrongly penalized. The inaccurate training signal makes NAT tend to generate a mixture of different translations rather than a consistent translation, which typically contains many repeated tokens in generated results.

Many efforts have been devoted to addressing the above problem (Libovický & Helcl, 2018; Shao et al., 2020; Ghazvininejad et al., 2020a; Du et al., 2021; Huang et al., 2022c). Among them, Directed Acyclic Transformer (DA-Transformer) (Huang et al., 2022c) introduces a directed acyclic graph (DAG) on top of the NAT decoder, where decoder hidden states are organized as a graph rather than a sequence. By modeling the dependency between vertices, DAG is able to capture multiple translation modalities simultaneously by assigning tokens in different translations to distinct vertices. In this way, DA-Transformer does not heavily rely on knowledge distillation (KD) (Kim & Rush, 2016; Zhou et al., 2020) to reduce training data modalities and can achieve superior performance on raw data.

Despite the success of DA-Transformer, training it with negative log-likelihood (NLL) loss, which marginalizes out the path from the joint distribution of DAG path and reference, is sub-optimal in the scenario of NAT. It implicitly introduces a strict monotonic alignment between reference tokens and vertices on all paths. Although DAG enables the model to capture different translations in different

---

*Corresponding author: Yang Feng. Contact: {mazhengrui21b,shaochenze18z,guishangtong21s, fengyang}@ict.ac.cn;zhangmin2021@hit.edu.cn.

[1]Source code: https://github.com/ictnlp/FA-DAT.

transition paths, only paths that are aligned verbatim with reference with a large probability will be well calibrated by NLL (See Section 2.3 for detailed analysis). It weakens DAG's ability to handle data multi-modality, making the model less confident in generating outputs and requiring a large graph size to achieve satisfying performance.

In this paper, we extend the verbatim alignment between reference and DAG path to a fuzzy alignment, aiming to better handle the multi-modality problem. Specifically, we do not require the exact alignment but hold the view that all paths in DAG are fuzzily aligned with the reference sentence. To indicate the quality of alignment, an alignment score is assigned to each DAG path based on the expectation of $n$-gram overlapping. We further define an alignment score between the whole DAG and reference as the expected alignment score of all its paths. The model is trained to maximize the alignment score, which takes captured translations in all modalities into account.

Experiments on major WMT benchmarks show that our method substantially improves the translation quality of DA-Transformer. It achieves comparable performance to the autoregressive Transformer without the help of knowledge distillation and beam search decoding, setting a new state of the art for NAT on the raw training data.

## 2 BACKGROUND

### 2.1 NON-AUTOREGRESSIVE MACHINE TRANSLATION

Non-autoregressive translation (Gu et al., 2018) is proposed to reduce the decoding latency. It abandons the assumption of autoregressive dependency between output tokens and generates all tokens simultaneously. Given a source sentence $\boldsymbol{x} = \{x_1, ..., x_N\}$, NAT factorizes the joint probability of target tokens $\boldsymbol{y} = \{y_1, ..., y_M\}$ as,

$$P_\theta(\boldsymbol{y}|\boldsymbol{x}) = \prod_i^M P_\theta(y_i|\boldsymbol{x}), \tag{1}$$

where $\theta$ is the model parameter and $P_\theta(y_i|\boldsymbol{x})$ denotes the translation probability of $y_i$ at position $i$.

In vanilla NAT, the decoder length is set to reference length during the training and determined by a trainable length predictor during the inference. Standard NAT model is trained with the cross-entropy loss, which strictly requires the generation of word $y_i$ at position $i$:

$$\mathcal{L}_{CE} = -\sum_i^M \log P_\theta(y_i|\boldsymbol{x}). \tag{2}$$

### 2.2 DIRECTED ACYCLIC TRANSFORMER

Vanilla NAT model suffers two major drawbacks, including inflexible length prediction and disability to handle multi-modality. DA-Transformer (Huang et al., 2022c) addresses these problems by stacking a directed acyclic graph on the top of NAT decoder, where hidden states and transitions between states represent vertices and edges in DAG.

Formally, given a bilingual pair $\boldsymbol{x} = \{x_1, ..., x_N\}$ and $\boldsymbol{y} = \{y_1, ..., y_M\}$, DA-Transformer sets the decoder length $L = \lambda \cdot N$ and models the translation probability by marginalizing out paths in DAG:

$$P_\theta(\boldsymbol{y}|\boldsymbol{x}) = \sum_{\boldsymbol{a} \in \Gamma_{\boldsymbol{y}}} P_\theta(\boldsymbol{y}|\boldsymbol{a}, \boldsymbol{x}) P_\theta(\boldsymbol{a}|\boldsymbol{x}), \tag{3}$$

where $\boldsymbol{a} = \{a_1, ..., a_M\}$ is a path represented by a sequence of vertex indexes with the bound $1 = a_1 < ... < a_M = L$ and $\Gamma_{\boldsymbol{y}}$ contains all paths with the same length as the target sentence $\boldsymbol{y}$. $P_\theta(\boldsymbol{a}|\boldsymbol{x})$ and $P_\theta(\boldsymbol{y}|\boldsymbol{a}, \boldsymbol{x})$ indicate the probability of path $\boldsymbol{a}$ and the probability of target sentence $\boldsymbol{y}$ conditioned on path $\boldsymbol{a}$ respectively. DAG factorizes the path probability $P_\theta(\boldsymbol{a}|\boldsymbol{x})$ based on the Markov hypothesis:

$$P_\theta(\boldsymbol{a}|\boldsymbol{x}) = \prod_{i=1}^{M-1} P_\theta(a_{i+1}|a_i, \boldsymbol{x}) = \prod_{i=1}^{M-1} \mathbf{E}_{a_i, a_{i+1}}, \tag{4}$$

where $\mathbf{E} \in \mathbb{R}^{L \times L}$ is a row-normalized transition matrix. Due to the unidirectional inherence of DAG, the lower triangular part of $\mathbf{E}$ are masked to zeros. Once path $\boldsymbol{a}$ is determined, token $y_i$ can be generated conditioned on the decoder hidden state with index $a_i$:

$$P_\theta(\boldsymbol{y}|\boldsymbol{a}, \boldsymbol{x}) = \prod_{i=1}^{M} P_\theta(y_i|a_i, \boldsymbol{x}). \tag{5}$$

DA-Transformer is trained to minimize the negative log-likelihood loss via dynamic programming:

$$\mathcal{L} = -\log P_\theta(\boldsymbol{y}|\boldsymbol{x}) = -\log \sum_{\boldsymbol{a} \in \Gamma_{\boldsymbol{y}}} P_\theta(\boldsymbol{y}|\boldsymbol{a}, \boldsymbol{x}) P_\theta(\boldsymbol{a}|\boldsymbol{x}). \tag{6}$$

The structure of DAG helps NAT model token dependency through transition probability between vertices while almost zero overhead of sequential operation is paid in decoding.

## 2.3 MULTI-MODALITY IN DAG

DA-Transformer alleviates the multi-modality problem by arranging tokens in different translations to distinct vertices, and the learned vertex transition probability prevents them from co-occurring in the same DAG path. Despite DAG's ability to capture different translations simultaneously, only paths that are aligned verbatim with reference with a large probability will be well calibrated by the NLL loss. It can be demonstrated by inspecting the gradients:[2]

$$\frac{\partial}{\partial \theta} \mathcal{L} = \sum_{\boldsymbol{a} \in \Gamma_y} P_\theta(\boldsymbol{a}|\boldsymbol{y}, \boldsymbol{x}) \frac{\partial}{\partial \theta} \mathcal{L}_{\boldsymbol{a}}, \tag{7}$$

where $\mathcal{L}_{\boldsymbol{a}} = -\log P_\theta(\boldsymbol{y}, \boldsymbol{a}|\boldsymbol{x})$. Equation 7 shows that NLL loss will assign paths with smaller posterior probability smaller gradient weights, making paths corresponding to translations in other modalities poorly calibrated. Considering that only one reference is provided in major translation benchmarks, training DAG with the NLL loss inevitably hurts its potential to handle translation multi-modality.

## 3 METHODOLOGY

In this section, we extend the verbatim alignment between reference and DAG path to an $n$-gram-based fuzzy alignment. In Section 3.1 and 3.2, we explore a metric to calibrate DAG under fuzzy alignment. We develop an efficient algorithm to calculate the alignment score in Section 3.3, and introduce a training strategy with fuzzy alignment in Section 3.4.

### 3.1 FUZZY ALIGNMENT

We do not require the exact alignment but hold the view that all paths in DAG are fuzzily aligned with the reference sentence to some extent. In order to allow multi-modality such as position shifts and word reorderings, we measure the quality of fuzzy alignment by the order-agnostic $n$-gram overlapping, which is a widely used factor in machine translation evaluation (Papineni et al., 2002).

Given a target sentence $\boldsymbol{y} = \{y_1, ..., y_M\}$ and an integer $n \geq 1$, we denote the set of all its non-repeating $n$-grams as $G_n(\boldsymbol{y})$, the elements of which are mutually exclusive. For any $n$-gram $\boldsymbol{g} \in G_n(\boldsymbol{y})$, the number of its appearances in $\boldsymbol{y}$ is denoted as $C_{\boldsymbol{g}}(\boldsymbol{y})$. We measure the alignment quality by clipped $n$-gram precision of model output $\boldsymbol{y}'$ against reference $\boldsymbol{y}$:

$$p_n(\boldsymbol{y}', \boldsymbol{y}) = \frac{\sum\limits_{\boldsymbol{g} \in G_n(\boldsymbol{y})} \min(C_{\boldsymbol{g}}(\boldsymbol{y}'), C_{\boldsymbol{g}}(\boldsymbol{y}))}{\sum\limits_{\boldsymbol{g} \in G_n(\boldsymbol{y}')} C_{\boldsymbol{g}}(\boldsymbol{y}')}. \tag{8}$$

Considering that the DAG path is composed of consecutive vertices, and each vertex represents a word distribution rather than a concrete word, we assign an alignment score to each DAG path with the expected $n$-gram precision:

$$p_n(\theta, \boldsymbol{a}, \boldsymbol{y}) = \mathbb{E}_{\boldsymbol{y}' \sim P_\theta(\boldsymbol{y}'|\boldsymbol{a}, \boldsymbol{x})}[p_n(\boldsymbol{y}', \boldsymbol{y})]. \tag{9}$$

---

[2]We leave the derivation in Appendix A.

### 3.2 Estimating Fuzzy Alignment in DAG

Directed acyclic graph simultaneously retains multiple translations in different paths, motivating us to measure the alignment between generated graph and reference sentence by the averaged alignment score of all its possible transition paths:

$$p_n(\theta, \boldsymbol{y}) = \mathbb{E}_{\boldsymbol{a} \sim P_\theta(\boldsymbol{a}|\boldsymbol{x})}[p_n(\theta, \boldsymbol{a}, \boldsymbol{y})]. \tag{10}$$

However, the search spaces for translations and paths are both exponentially large, making Equation 10 intractable. Alternatively, we turn to calculate the ratio of the clipped expected count of $n$-gram matching to expected number of $n$-grams, which can be considered as an approximation of $p_n(\theta, \boldsymbol{y})$:

$$p'_n(\theta, \boldsymbol{y}) = \frac{\sum\limits_{\boldsymbol{g} \in G_n(\boldsymbol{y})} \min(\mathbb{E}_{\boldsymbol{y}'}[C_{\boldsymbol{g}}(\boldsymbol{y}')], C_{\boldsymbol{g}}(\boldsymbol{y}))}{\mathbb{E}_{\boldsymbol{y}'}[\sum\limits_{\boldsymbol{g} \in G_n(\boldsymbol{y}')} C_{\boldsymbol{g}}(\boldsymbol{y}')]}. \tag{11}$$

Though $p'_n(\theta, \boldsymbol{y})$ is much simplified, the direct calculation will still suffer from exponential time complexity due to the intractable expectation terms in it. In the following section, we will focus on developing an efficient algorithm to calculate $\mathbb{E}_{\boldsymbol{y}'}[C_{\boldsymbol{g}}(\boldsymbol{y}')]$ and $\mathbb{E}_{\boldsymbol{y}'}[\sum\limits_{\boldsymbol{g} \in G_n(\boldsymbol{y}')} C_{\boldsymbol{g}}(\boldsymbol{y}')]$.

### 3.3 Efficient Calculation of Fuzzy Alignment

Fortunately, the Markov property of DAG makes it possible to simplify the calculation of Equation 11. We first consider an $n$-vertex subpath $v_1, ..., v_n$, which is bounded by $1 \leq v_1, ..., v_n \leq L$.[3] An indicator function $\mathbb{1}(v_1, ..., v_n \in \boldsymbol{a})$ is introduced to indicate whether $v_1, ..., v_n$ is a part of path $\boldsymbol{a}$. Then we can define the passing probability of a subpath by enumerating all possible paths $\boldsymbol{a}$:

$$P(\mathbb{1}(v_1, ..., v_n)|\boldsymbol{x}) = \sum_{\boldsymbol{a}} P_\theta(\boldsymbol{a}|\boldsymbol{x})\mathbb{1}(v_1, ..., v_n \in \boldsymbol{a}). \tag{12}$$

With the definition of passing probability, we can transform the sum operations in Equation 11 from enumerating all translations to enumerating all the $n$-vertex subpaths, which significantly narrows the search space. We directly give the following theorem and refer readers to Appendix B for detailed derivation:

$$\mathbb{E}_{\boldsymbol{y}'}[\sum_{\boldsymbol{g} \in G_n(\boldsymbol{y}')} C_{\boldsymbol{g}}(\boldsymbol{y}')] = \sum_{v_1, ..., v_n} P(\mathbb{1}(v_1, ..., v_n)|\boldsymbol{x}), \tag{13}$$

$$\mathbb{E}_{\boldsymbol{y}'}[C_{\boldsymbol{g}}(\boldsymbol{y}')] = \sum_{v_1, ..., v_n} P(\mathbb{1}(v_1, ..., v_n)|\boldsymbol{x}) \prod_{i=1}^{n} P_\theta(\boldsymbol{g}_i|v_i), \tag{14}$$

where $P_\theta(\boldsymbol{g}_i|v_i)$ denotes the probability that vertex with index $v_i$ generates the $i$-th token in $n$-gram $\boldsymbol{g}$. By narrowing the search space, we can find that those expected counts are actually the sum and weighted sum of passing probabilities of $n$-vertex subpaths. It is worth noting that the $n$-vertex subpath $v_1, ..., v_n$ forms a Markov chain, leading us to factorize the passing probability into the product of transitions. We reuse the transition matrix $\mathbf{E}$ introduced in Section 2.2 for notation, where $\mathbf{E}_{v_i, v_j}$ is the transition probability from $v_i$ to $v_j$. The passing probability of an $n$-vertex subpath can be formulated as:

$$P(\mathbb{1}(v_1, ..., v_n)|\boldsymbol{x}) = P(\mathbb{1}(v_1)|\boldsymbol{x}) \prod_{i=1}^{n-1} \mathbf{E}_{v_i, v_{i+1}}, \tag{15}$$

where $P(\mathbb{1}(v)|\boldsymbol{x})$ denotes the passing probability of vertex with index $v$. For any $1 \leq v \leq L$, $P(\mathbb{1}(v)|\boldsymbol{x})$ can be calculated efficiently via dynamic programming:

$$P(\mathbb{1}(v)|\boldsymbol{x}) = \sum_{v' < v} P(\mathbb{1}(v')|\boldsymbol{x})\mathbf{E}_{v', v}, \tag{16}$$

---

[3]Note that there is no need to require $1 = v_1 < ... < v_n = L$. In fact, the upper triangular transition matrix only considers paths with monotonic order and sets the probability of others to 0.

---

**Algorithm 1** Calculation of $\mathbb{E}_{\boldsymbol{y}'}[C_{\boldsymbol{g}}(\boldsymbol{y}')]$ and $\mathbb{E}_{\boldsymbol{y}'}[\sum_{\boldsymbol{g} \in G_n(\boldsymbol{y}')} C_{\boldsymbol{g}}(\boldsymbol{y}')]$

---

**Input:** Transition matrix $\mathbf{E}$, Token probability matrix $\mathbf{G}$, Graph size $L$, $n$-gram order $N$.
**Output:** Expected counts $\mathbb{E}_{\boldsymbol{y}'}[C_{\boldsymbol{g}}(\boldsymbol{y}')]$ and $\mathbb{E}_{\boldsymbol{y}'}[\sum_{\boldsymbol{g} \in G_n(\boldsymbol{y}')} C_{\boldsymbol{g}}(\boldsymbol{y}')]$.
1: $p_1 \leftarrow 1$
2: **for** $i \leftarrow 2$ to $L$ **do**
3: $\quad p_i \leftarrow \sum_{j=1}^{i-1} p_j \mathbf{E}_{j,i}$
4: $\boldsymbol{p} \leftarrow [p_1, p_2, ..., p_L]^T$
5: $\boldsymbol{c}^{(1)} \leftarrow \boldsymbol{p}^T; \boldsymbol{c}^{(2)} \leftarrow \boldsymbol{p}^T \odot \mathbf{G}_{1,:}$
6: **for** $i \leftarrow 2$ to $N$ **do**
7: $\quad \boldsymbol{c}^{(1)} \leftarrow \boldsymbol{c}^{(1)} \cdot \mathbf{E}$
8: $\quad \boldsymbol{c}^{(2)} \leftarrow (\boldsymbol{c}^{(2)} \cdot \mathbf{E}) \odot \mathbf{G}_{i,:}$
9: $\mathbb{E}_{\boldsymbol{y}'}[C_{\boldsymbol{g}}(\boldsymbol{y}')] \leftarrow \|\boldsymbol{c}^{(2)}\|_1; \mathbb{E}_{\boldsymbol{y}'}[\sum_{\boldsymbol{g} \in G_n(\boldsymbol{y}')} C_{\boldsymbol{g}}(\boldsymbol{y}')] \leftarrow \|\boldsymbol{c}^{(1)}\|_1$
10: **return** $\mathbb{E}_{\boldsymbol{y}'}[C_{\boldsymbol{g}}(\boldsymbol{y}')]$ and $\mathbb{E}_{\boldsymbol{y}'}[\sum_{\boldsymbol{g} \in G_n(\boldsymbol{y}')} C_{\boldsymbol{g}}(\boldsymbol{y}')]$.

---

with the boundary condition that the passing probability of the first vertex in the directed graph is equal to 1. For convenience, we introduce a passing probability vector $\boldsymbol{p} \in \mathbb{R}^L$, where $\boldsymbol{p}_v = P(\mathbb{1}(v)|\boldsymbol{x})$. Equation 15 reminds us that the sum of $n$-vertex passing probabilities is actually the sum of $(n-1)$-hop transition probabilities in a Markov chain with initial probability $\boldsymbol{p}$ and transition matrix $\mathbf{E}$. On the basis of Equation 13, we have:

$$\mathbb{E}_{\boldsymbol{y}'}[\sum_{\boldsymbol{g} \in G_n(\boldsymbol{y}')} C_{\boldsymbol{g}}(\boldsymbol{y}')] = \sum_{v_1,...,v_n} P(\mathbb{1}(v_1)|\boldsymbol{x}) \prod_{i=1}^{n-1} \mathbf{E}_{v_i,v_{i+1}} = \|\boldsymbol{p}^T \cdot \mathbf{E}^{n-1}\|_1. \tag{17}$$

By rearranging the order of products, we can handle the weighted sum of passing probabilities in Equation 14 in a similar way. We introduce $\mathbf{G} \in \mathbb{R}^{n \times L}$, with the $i$th row of $\mathbf{G}$ representing the token probability of $\boldsymbol{g}_i$ at each vertex, i.e., $\mathbf{G}_{i,v} = P_\theta(\boldsymbol{g}_i|v)$. Then we can obtain:

$$\begin{aligned}
\mathbb{E}_{\boldsymbol{y}'}[C_{\boldsymbol{g}}(\boldsymbol{y}')] &= \sum_{v_1,...,v_n} P(\mathbb{1}(v_1)|\boldsymbol{x}) \prod_{i=1}^{n-1} \mathbf{E}_{v_i,v_{i+1}} \prod_{i=1}^{n} P_\theta(\boldsymbol{g}_i|v_i) \\
&= \sum_{v_1,...,v_n} (P(\mathbb{1}(v_1)|\boldsymbol{x}) P_\theta(\boldsymbol{g}_1|v_1)) \prod_{i=1}^{n-1} (\mathbf{E}_{v_i,v_{i+1}} P_\theta(\boldsymbol{g}_{i+1}|v_{i+1})) \\
&= \|\boldsymbol{p}^T \odot \mathbf{G}_{1,:} \cdot \mathbf{E} \odot \mathbf{G}_{2,:} \cdot ... \cdot \mathbf{E} \odot \mathbf{G}_{n,:}\|_1,
\end{aligned} \tag{18}$$

where $\odot$ denotes position-wise product, and $\mathbf{G}_{i,:}$ should be broadcast when $i > 1$. We refer readers to Appendix C for detailed derivation.

Based on the discussion above, we finally avoid the exponential time complexity in calculating $p_n'(\theta, \boldsymbol{y})$. Instead, it can be implemented with $\mathcal{O}(n + L)$ parallel operations, making the proposed fuzzy alignment objective practical for the training. We summarize the calculation process in Algorithm 1.

### 3.4 TRAINING STRATEGY

We have found a way to train DA-Transformer to maximize the fuzzy alignment score $p_n'(\theta, \boldsymbol{y})$ efficiently. However, it may bias the model to favor short translations. To this end, we adopt the idea of *brief penalty* (Papineni et al., 2002) for regularization. Specifically, we define the brief penalty of DAG based on the ratio of reference length to the expected translation length:

$$BP = \min(\exp{(1 - \frac{T_{\boldsymbol{y}}}{\mathbb{E}_{\boldsymbol{y}'}[T_{\boldsymbol{y}'}]})}, 1), \tag{19}$$

where $T_{\boldsymbol{y}}$ denotes the length of reference $\boldsymbol{y}$. As the expected translation length is equal to the expected number of 1-grams in outputs, it is convenient to calculate $\mathbb{E}_{\boldsymbol{y}'}[T_{\boldsymbol{y}'}]$ efficiently with the help of Equation 17. Finally, we train the model with the following loss:

$$\mathcal{L} = -BP \times p_n'(\theta, \boldsymbol{y}). \tag{20}$$

We first pretrain DA-Transformer with the NLL loss to obtain a good initialization, and then finetune the model with our fuzzy alignment objective.

## 4 EXPERIMENTS

### 4.1 EXPERIMENTAL SETUP

**Datasets** We conduct experiments on two major benchmarks that are widely used in previous studies: WMT14 English↔German (EN↔DE, 4M) and WMT17 Chinese↔English (ZH↔EN, 20M).[4] We apply BPE (Sennrich et al., 2016) to learn a joint subword vocabulary for EN↔DE and separate vocabularies for ZH↔EN on the tokenized data. For fair comparison, we evaluate our method with SacreBLEU (Post, 2018)[5] for WMT EN-ZH task and tokenized BLEU (Papineni et al., 2002) for other benchmarks. Considering BLEU may be biased, we also measure the translation quality with learned metrics in Appendix F. The decoding speedup is measured with a batch size of 1.

**Implementation Details** We adopt Transformer-*base* (Vaswani et al., 2017) as our autoregressive baseline. For the architecture of our model, we strictly follow the settings of DA-Transformer, where we set the decoder length to 8 times the source length ($\lambda = 8$) and use graph positional embeddings as decoder inputs unless otherwise specified. During both pretraining and finetuning, we set dropout rate to 0.1, weight decay to 0.01, and no label smoothing is applied. In pretraining, all models are trained for 300k updates with a batch size of 64k tokens. The learning rate warms up to $5 \cdot 10^{-4}$ within 10k steps. In finetuning, we use the batch of 256k tokens to stabilize the gradients and train models for 5k updates. The learning rate warms up to $2 \cdot 10^{-4}$ within 500 steps. We evaluate BLEU scores on the validation set and average the best 5 checkpoints for the final model. We implement our models with open-source toolkit `fairseq` (Ott et al., 2019). All the experiments are conducted on GeForce RTX 3090 GPUs.

**Glancing** Glancing (Qian et al., 2021a) is a promising strategy to alleviate data multi-modality while training. We apply the same glancing strategy as Huang et al. (2022c), which assigns target tokens to appropriate vertices based on the most probable path: $\widetilde{\boldsymbol{a}} = \arg\max_{\boldsymbol{a}} P_\theta(\boldsymbol{a}|\boldsymbol{x}, \boldsymbol{y})$. In the pretraining, we linearly anneal the unmasking ratio $\tau$ from 0.5 to 0.1. In the finetuning, we fix $\tau$ to 0.1.

**Decoding** We apply beam search with a beam size of 5 for autoregressive baseline and argmax decoding for vanilla NAT. Following Huang et al. (2022c), we find the translation of DAG with *Greedy* and *Lookahead* decoding. The former only searches the path greedily and then collects the most probable token from each vertex sequentially:

$$a_i^* = \arg\max_{a_i} P_\theta(a_i|a_{i-1}, \boldsymbol{x}), \; y_i^* = \arg\max_{y_i} P_\theta(y_i|a_i, \boldsymbol{x}). \tag{21}$$

And the latter jointly searches the path and tokens in a greedy way:

$$a_i^*, y_i^* = \arg\max_{a_i, y_i} P_\theta(y_i|a_i, \boldsymbol{x}) P_\theta(a_i|a_{i-1}, \boldsymbol{x}). \tag{22}$$

We also adopt *Joint-Viterbi* decoding (Shao et al., 2022a) to find the global joint optimum of translation and path under pre-defined length constraint, then rerank those candidates by length normalization.

### 4.2 PRELIMINARY EXPERIMENT: EFFECTS OF $n$

We first study the effects of $n$-gram order on our fuzzy alignment objective. In this experiment, we set the decoder length to 4 times the source length ($\lambda = 4$) and apply Lookahead decoding to generate outputs. We compare different settings of $n$ and report BLEU scores and BERTScores on the raw WMT14 EN-DE dataset in Table 1.

As shown in Table 1, the performance of our fuzzy alignment objective differs as $n$ varies. We find that using a larger $n$ is helpful for DAG to generate better translations. Specifically, it improves the DA-Transformer baseline by 0.82 BLEU and 1.11 BERTScore when $n = 2$ and by 0.65 BLEU and

---

[4]*Newstest2013* as the validation set and *newstest2014* as the test set for EN↔DE; *devtest2017* as the validation set and *newstest2017* as the test set for ZH↔EN.

[5]SacreBLEU signature: BLEU+case.mixed+numrefs.1+smooth.exp+tok.zh+version.1.5.1

Table 1: Effects of $n$ on our fuzzy alignment objective on **raw** WMT14 EN-DE. Decoder length is set to 4 times source length ($\lambda = 4$).

|  | BLEU | BERTScore |
|---|---|---|
| DA-Transformer | 26.16 | 60.57 |
| +1-gram | 25.71 | 60.69 |
| +2-gram | 26.98 | 61.68 |
| +3-gram | 26.81 | 61.56 |

Table 2: BLEU scores on **raw** WMT14 EN↔DE and WMT17 ZH↔EN dataset. Results of baselines are quoted from Huang et al. (2022c). [†] indicates the results from our re-implementation. * and ** indicate the improvement over DA-Transformer is statistically significant ($p < 0.05$ and $p < 0.01$, respectively).

| Model | | Iter. | Speedup | WMT14 EN-DE | WMT14 DE-EN | WMT17 ZH-EN | WMT17 EN-ZH |
|---|---|---|---|---|---|---|---|
| Transformer (Vaswani et al., 2017) | | N | 1.0× | 27.6 | 31.4 | 23.7 | 34.3 |
| Transformer[†] | | N | 1.0× | 27.54 | 31.55 | 24.23 | 35.19 |
| CMLM (Ghazvininejad et al., 2019) | | 10 | 2.2× | 24.61 | 29.40 | - | - |
| SMART (Ghazvininejad et al., 2020b) | | 10 | 2.2× | 25.10 | 29.58 | - | - |
| DisCo (Kasai et al., 2020) | | ≈4 | 3.5× | 25.64 | - | - | - |
| Imputer (Saharia et al., 2020) | | 8 | 2.7× | 25.0 | - | - | - |
| CMLMC (Huang et al., 2022d) | | 10 | 1.7× | 26.40 | 30.92 | - | - |
| Vanilla-NAT (Gu et al., 2018) | | 1 | 15.3× | 11.79 | 16.27 | 8.69 | 18.92 |
| CTC (Libovický & Helcl, 2018) | | 1 | 14.6× | 18.42 | 23.65 | 12.23 | 26.84 |
| AXE (Ghazvininejad et al., 2020a) | | 1 | 14.2× | 20.40 | 24.90 | - | - |
| GLAT (Qian et al., 2021a) | | 1 | 15.3× | 19.42 | 26.51 | 18.88 | 29.79 |
| OaXE (Du et al., 2021) | | 1 | 14.2× | 22.4 | 26.8 | - | - |
| CTC + GLAT (Qian et al., 2021b) | | 1 | 14.6× | 25.02 | 29.14 | 19.92 | 30.65 |
| CTC + DSLP (Huang et al., 2022a) | | 1 | 14.0× | 24.81 | 28.33 | - | - |
| DA-Transformer | + *Greedy* | 1 | 14.2× | 26.08 | 30.48 | 22.66 | 33.27 |
| (Huang et al., 2022c) | + *Lookahead* | 1 | 14.0× | 26.57 | 30.68 | 22.82 | 33.83 |
| | + *Greedy* | 1 | 14.2× | 26.07 | 30.69 | 22.35 | 33.58 |
| DA-Transformer[†] | + *Lookahead* | 1 | 14.0× | 26.56 | 30.81 | 22.65 | 33.62 |
| | + *Joint-Viterbi* | 1 | 13.2× | 26.89 | 31.09 | 23.17 | 33.25 |
| | + *Greedy* | 1 | 14.2× | 27.49** | 31.36** | 23.78** | 33.97* |
| FA-DAT | + *Lookahead* | 1 | 14.0× | **27.53**** | 31.37** | 23.81** | 34.02** |
| | + *Joint-Viterbi* | 1 | 13.2× | 27.47** | **31.44*** | **24.22**** | **34.49**** |

0.99 BERTScore when $n = 3$. We speculate the reason for performance degradation with $n = 1$ is that the 1-gram alignment objective thoroughly breaks the order dependency among tokens in a sentence, which will encourage the model to output bag-of-words instead of a fluent sentence. It is also noteworthy that 2-gram alignment performs better than 3-gram. We attribute the reason to the Markov property of DAG, which only models the dependency between adjacent tokens, thus making the 2-gram alignment more appropriate. In the following experiments, we will apply $n = 2$ as the default setting and name it as **FUZZY-ALIGNED DIRECTED ACYCLIC TRANSFORMER** (**FA-DAT**).

## 4.3 MAIN RESULTS

We compare our FA-DAT with the autoregressive baseline and previous NAT approaches in Table 2. FA-DAT consistently improves the translation quality of DA-Transformer by a large margin under all decoding strategies. Notably, it achieves comparable performance with the autoregressive baseline without knowledge distillation and beam search decoding. With Joint-Viterbi decoding, the average gap between autoregressive model and FA-DAT on raw data is further reduced to 0.09 BLEU on EN↔DE and 0.35 BLEU on ZH↔EN, while FA-DAT maintains 13.2 times decoding speedup.

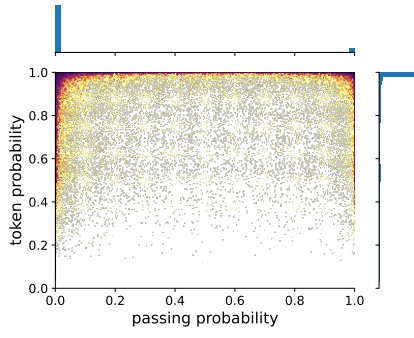

(a) Vertex distribution of FA-DAT

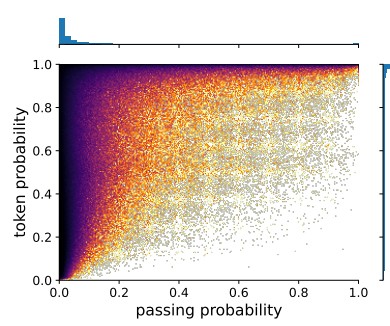

(b) Vertex distribution of DA-Transformer

Figure 1: The distribution of vertices' passing probabilities and max token probabilities on test set of WMT14 EN-DE. Passing probability and max token probability refer to the probability of a vertex appearing on a sampled path and the probability of its most probable token respectively. Darker area indicates dense distribution of vertices. Marginal distributions are given on the top and right.

Table 3: Statistics of translations on **raw** WMT14 EN-DE dataset.

|  |  | $-\log P_\theta(\boldsymbol{a}|\boldsymbol{x})$ | $-\log P_\theta(\boldsymbol{y}|\boldsymbol{a}, \boldsymbol{x})$ | $-\log P_\theta(\boldsymbol{y}|\boldsymbol{x})$ |
|---|---|---|---|---|
| DA-Transformer | +*Greedy* | 10.27 | 4.51 | 11.47 |
|  | +*Lookahead* | 10.49 | 3.84 | 11.06 |
|  | +*Joint-Viterbi* | 9.60 | 3.32 | 10.50 |
| FA-DAT | +*Greedy* | 2.99 | 1.03 | 2.03 |
|  | +*Lookahead* | 3.00 | 1.00 | 2.02 |
|  | +*Joint-Viterbi* | 3.17 | 0.98 | 1.86 |

## 4.4 ANALYSIS AND DISCUSSION

**Generation Confidence** We note that the performance of Greedy decoding is on par with that of Lookahead decoding in FA-DAT while left far behind in DA-Transformer in Table 2. To conduct the analysis, we collect the probability of generated path ($P_\theta(\boldsymbol{a}|\boldsymbol{x})$) and generated translation given path ($P_\theta(\boldsymbol{y}|\boldsymbol{a}, \boldsymbol{x})$) on test set and further calculate the marginal probability of generated translation ($P_\theta(\boldsymbol{y}|\boldsymbol{x})$) via dynamic programming. Statistics are shown in Table 3. We observe $-\log P_\theta(\boldsymbol{y}|\boldsymbol{a}, \boldsymbol{x})$ is extremely low in FA-DAT, suggesting that every vertex in DAG is assigned with a concrete token with high confidence, making Greedy decoding perform similarly to Lookahead decoding. Compared with DA-Transformer, path searched in FA-DAT has a larger probability and the model is more confident about its output translation under all decoding strategies. We owe it to FA-DAT's ability to well calibrate vertices in different modalities. It can be verified by checking the distribution of vertices' passing

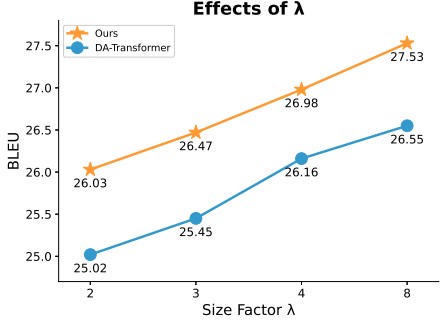

Figure 2: Effects of $\lambda$ on **raw** WMT14 EN-DE. Decoder length is set to $\lambda$ times source length.

probabilities and max token probabilities in Figure 1. Vertices of FA-DAT have passing probabilities either close to 0 or to 1 and token probabilities all close to 1. It demonstrates the fuzzy alignment objective's ability to reduce model perplexity and improve generation confidence.

**Effects of Graph Size** We further study how the size of the graph affects our method. We vary the graph size hyperparameter $\lambda$ from 2 to 8 and measure the translation quality of the proposed

FA-DAT and DA-Transformer baseline with Lookahead decoding on WMT14 EN-DE. As shown in Figure 2, FA-DAT consistently outperforms DA-Transformer by 0.96 BLEU on average across all settings of graph size. Notably, FA-DAT does not necessarily require a large graph size to perform competitively. It makes use of vertices in DAG more efficiently, realizing comparable performance with $\lambda = 3$ to DA-Transformer with $\lambda = 8$ (26.47 vs 26.55). We attribute the success to that fuzzy alignment considers all translation modalities during training. Vertices corresponding to tokens in different modalities are all incorporated in the backward flow with non-negligible gradients. Oppositely, NLL loss only trains paths that are strictly aligned with the reference. As shown in Figure 1, most vertices' token probabilities are close to 1 in FA-DAT while are scattered in DA-Transformer. It suggests all the vertices in DAG are well calibrated with the fuzzy alignment objective. Since model complexity is quadratic to graph size, massive overhead will be introduced with a larger graph size. FA-DAT achieves a better trade-off between graph size and translation quality, making the structure of DAG more appealing in practical application.

## 5 RELATED WORK

Gu et al. (2018) first proposes non-autoregressive translation to reduce the decoding latency. Compared with the autoregressive counterpart, NAT suffers from the performance degradation due to the multi-modality problem and heavily relies on knowledge distilled from autoregressive teacher (Kim & Rush, 2016; Zhou et al., 2020; Tu et al., 2020; Ding et al., 2021; Shao et al., 2022b). The multi-modality problem in training data has been analyzed comprehensively in the view of lexical choice (Ding et al., 2021), syntactic structure (Zhang et al., 2022) and information theory (Huang et al., 2022b). To mitigate the performance gap, some researchers introduce semi-autoregressive decoding (Wang et al., 2018; Ran et al., 2020; Ghazvininejad et al., 2020b; Wang et al., 2021) or iterative decoding mechanisms to refine NAT outputs (Lee et al., 2018; Ghazvininejad et al., 2019; Gu et al., 2019; Kasai et al., 2020; Saharia et al., 2020; Huang et al., 2022d). However, these approaches inevitably weaken the advantage of fast inference (Kasai et al., 2021). To this end, some researchers focus on improving the training procedure of fully NAT models. Qian et al. (2021a) introduces glancing training to NAT, which helps the model to calibrate outputs by feeding partial targets. Huang et al. (2022a) further applies a similar idea to middle layers with deep supervision. In another direction, researchers are looking for flexible training objectives that alleviate strict position-wise alignment required by the naive NLL loss. Libovický & Helcl (2018) proposes latent alignment model with CTC loss (Graves et al., 2006) and Shao & Feng (2022) further explores non-monotonic alignments under CTC loss. Wang et al. (2019) introduces two auxiliary regularization terms to improve the quality of decoder hidden representations. Shao et al. (2020; 2021) introduce sequence-level training objectives with reinforcement learning and bag-of-ngrams difference. Ghazvininejad et al. (2020a) trains NAT model with the best monotonic alignment found by dynamic programming and Du et al. (2021) further extends it to order-agnostic cross-entropy loss. Recently, Huang et al. (2022c) introduces DA-Transformer, which alleviates multi-modality problem by modeling dependency among vertices in directed acyclic graph. Despite the success of DA-Transformer, it still implicitly requires monotonic one-to-one alignment between target tokens and vertices in the training, which weakens its ability to handle multi-modality.

## 6 CONCLUSION

In this paper, we introduce a fuzzy alignment objective between the directed acyclic graph and reference sentence based on $n$-gram matching. Our proposed objective can better handle predictions with position shift, word reordering, or length variation, which are critical sources of translation multi-modality. Experiments demonstrate that our method facilitates training of DA-Transformer, achieves comparable performance to autoregressive baseline with fully parallel decoding, and sets new state of the art for NAT on the raw training data.

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

## A    PROOF OF EQUATION 7

We can inspect the NLL gradients from different translation paths in the graph by applying the chain rule:

$$
\begin{aligned}
\frac{\partial}{\partial \theta}\mathcal{L} &= \frac{\partial}{\partial \theta}(-\log P_\theta(\boldsymbol{y}|\boldsymbol{x})) \\
&= -\frac{1}{P_\theta(\boldsymbol{y}|\boldsymbol{x})}\frac{\partial}{\partial \theta}P_\theta(\boldsymbol{y}|\boldsymbol{x}) \\
&= -\frac{1}{P_\theta(\boldsymbol{y}|\boldsymbol{x})}\frac{\partial}{\partial \theta}\Big(\sum_{\boldsymbol{a}\in\Gamma_{\boldsymbol{y}}}P_\theta(\boldsymbol{y},\boldsymbol{a}|\boldsymbol{x})\Big) \\
&= -\frac{1}{P_\theta(\boldsymbol{y}|\boldsymbol{x})}\sum_{\boldsymbol{a}\in\Gamma_{\boldsymbol{y}}}\frac{\partial}{\partial \theta}P_\theta(\boldsymbol{y},\boldsymbol{a}|\boldsymbol{x}) \\
&= \sum_{\boldsymbol{a}\in\Gamma_{\boldsymbol{y}}}\Big(\frac{P_\theta(\boldsymbol{y},\boldsymbol{a}|\boldsymbol{x})}{P_\theta(\boldsymbol{y}|\boldsymbol{x})}\Big)\Big(-\frac{1}{P_\theta(\boldsymbol{y},\boldsymbol{a}|\boldsymbol{x})}\frac{\partial}{\partial \theta}P_\theta(\boldsymbol{y},\boldsymbol{a}|\boldsymbol{x})\Big) \\
&= \sum_{\boldsymbol{a}\in\Gamma_{\boldsymbol{y}}}P_\theta(\boldsymbol{a}|\boldsymbol{y},\boldsymbol{x})\frac{\partial}{\partial \theta}(-\log P_\theta(\boldsymbol{y},\boldsymbol{a}|\boldsymbol{x}))
\end{aligned}
\tag{23}
$$

Note that $-\log P_\theta(\boldsymbol{y},\boldsymbol{a}|\boldsymbol{x})$ is the loss of one specific translation path. NLL implicitly assigns each path its posterior probability $P_\theta(\boldsymbol{a}|\boldsymbol{y},\boldsymbol{x})$ as the weight during training, making paths corresponding to translations in other modalities poorly calibrated.

## B    PROOF OF EQUATION 13 AND EQUATION 14

We denote the length of arbitrary translation $\boldsymbol{y}'$ and that of path $\boldsymbol{a}$ as $T_{\boldsymbol{y}'}$ and $T_{\boldsymbol{a}}$ respectively. Note that the averaged length of generated translations should be the same as the averaged length of DAG paths:

$$
\sum_{\boldsymbol{y}'}P_\theta(\boldsymbol{y}'|\boldsymbol{x})T_{\boldsymbol{y}'} = \sum_{\boldsymbol{a}}P_\theta(\boldsymbol{a}|\boldsymbol{x})T_{\boldsymbol{a}}
\tag{24}
$$

Then we have:

$$
\begin{aligned}
\mathbb{E}_{\boldsymbol{y}'}\Big[\sum_{\boldsymbol{g}\in G_n(\boldsymbol{y}')}C_{\boldsymbol{g}}(\boldsymbol{y}')\Big] &= \sum_{\boldsymbol{y}'}P_\theta(\boldsymbol{y}'|\boldsymbol{x})\sum_{\boldsymbol{g}\in G_n(\boldsymbol{y}')}C_{\boldsymbol{g}}(\boldsymbol{y}') \\
&= \sum_{\boldsymbol{y}'}P_\theta(\boldsymbol{y}'|\boldsymbol{x})(T_{\boldsymbol{y}'}-n+1) \\
&= \sum_{\boldsymbol{a}}P_\theta(\boldsymbol{a}|\boldsymbol{x})(T_{\boldsymbol{a}}-n+1) \\
&= \sum_{\boldsymbol{a}}P_\theta(\boldsymbol{a}|\boldsymbol{x})\sum_{v_1,\dots,v_n}\mathbb{1}(v_1,\dots,v_n\in\boldsymbol{a}) \\
&= \sum_{v_1,\dots,v_n}P(\mathbb{1}(v_1,\dots,v_n)|\boldsymbol{x})
\end{aligned}
\tag{25}
$$

We can handle $\mathbb{E}_{\boldsymbol{y}'}[C_{\boldsymbol{g}}(\boldsymbol{y}')]$ in a similar way. When a particular path $\boldsymbol{a}$ is given, it is convenient to calculate the averaged appearance of $n$-gram $\boldsymbol{g}$ through sliding a window of size $n$ on token distributions along the path (Shao et al., 2020):

$$
\mathbb{E}_{\boldsymbol{y}'|\boldsymbol{a}}[C_{\boldsymbol{g}}(\boldsymbol{y}')] = \sum_{\boldsymbol{y}'}P_\theta(\boldsymbol{y}'|\boldsymbol{a},\boldsymbol{x})C_{\boldsymbol{g}}(\boldsymbol{y}') = \sum_{i=1}^{T_a-n+1}\prod_{j=1}^{n}P_\theta(\boldsymbol{g}_j|a_{i+j-1})
\tag{26}
$$

Then we apply the indicator function $\mathbb{1}(v_1, ..., v_n \in \boldsymbol{a})$ to transform the enumerating space from the space of $n$ adjacent vertices in path $\boldsymbol{a}$ to the space of arbitrary $n$ vertices in graph:

$$
\begin{aligned}
\mathbb{E}_{\boldsymbol{y}'}[C_{\boldsymbol{g}}(\boldsymbol{y}')] &= \sum_{\boldsymbol{a}} P_\theta(\boldsymbol{a}|\boldsymbol{x}) \mathbb{E}_{\boldsymbol{y}'|\boldsymbol{a}}[C_{\boldsymbol{g}}(\boldsymbol{y}')] \\
&= \sum_{\boldsymbol{a}} P_\theta(\boldsymbol{a}|\boldsymbol{x}) \sum_{i=1}^{T_a-n+1} \prod_{j=1}^{n} P_\theta(\boldsymbol{g}_j|a_{i+j-1}) \\
&= \sum_{\boldsymbol{a}} P_\theta(\boldsymbol{a}|\boldsymbol{x}) \sum_{v_1,...,v_n} \mathbb{1}(v_1, ..., v_n \in \boldsymbol{a}) \prod_{j=1}^{n} P_\theta(\boldsymbol{g}_j|v_j) \\
&= \sum_{v_1,...,v_n} \sum_{\boldsymbol{a}} P_\theta(\boldsymbol{a}|\boldsymbol{x}) \mathbb{1}(v_1, ..., v_n \in \boldsymbol{a}) \prod_{j=1}^{n} P_\theta(\boldsymbol{g}_j|v_j) \\
&= \sum_{v_1,...,v_n} [\sum_{\boldsymbol{a}} P_\theta(\boldsymbol{a}|\boldsymbol{x}) \mathbb{1}(v_1, ..., v_n \in \boldsymbol{a})] \prod_{j=1}^{n} P_\theta(\boldsymbol{g}_j|v_j) \\
&= \sum_{v_1,...,v_n} P(\mathbb{1}(v_1, ..., v_n)) \prod_{j=1}^{n} P_\theta(\boldsymbol{g}_j|v_j)
\end{aligned}
\tag{27}
$$

We can find $\mathbb{E}_{\boldsymbol{y}'}[C_{\boldsymbol{g}}(\boldsymbol{y}')]$ is actually a weighted sum of passing probabilities of $n$-vertex subpaths.

## C    PROOF OF EQUATION 18

On the basis of Equation 14 and Equation 15, we first rewrite it as:

$$
\mathbb{E}_{\boldsymbol{y}'}[C_{\boldsymbol{g}}(\boldsymbol{y}')] = \sum_{v_1,...,v_n} P(\mathbb{1}(v_1)|\boldsymbol{x}) \prod_{i=1}^{n-1} \mathbf{E}_{v_i,v_{i+1}} \prod_{i=1}^{n} P_\theta(\boldsymbol{g}_i|v_i)
\tag{28}
$$

We can construct a similar form to Equation 17 by commutating and associating terms in the product:

$$
\begin{aligned}
\mathbb{E}_{\boldsymbol{y}'}[C_{\boldsymbol{g}}(\boldsymbol{y}')] &= \sum_{v_1,...,v_n} P(\mathbb{1}(v_1)|\boldsymbol{x}) \prod_{i=1}^{n-1} \mathbf{E}_{v_i,v_{i+1}} \prod_{i=1}^{n} P_\theta(\boldsymbol{g}_i|v_i) \\
&= \sum_{v_1,...,v_n} [P(\mathbb{1}(v_1)|\boldsymbol{x})P_\theta(\boldsymbol{g}_1|v_1)] \prod_{i=1}^{n-1} [\mathbf{E}_{v_i,v_{i+1}} P_\theta(\boldsymbol{g}_{i+1}|v_{i+1})]
\end{aligned}
\tag{29}
$$

Equation 29 indicates that the weighted sum of passing probability shares the same form with the sum of $(n-1)$-hop transition probabilities of a Markov chain with unnormalized initial probability and step-variant transition matrix. The initial probability and the $i$th step transition probability are given as:

$$
\begin{cases}
\tilde{P}(\mathbb{1}(v)|\boldsymbol{x}) = P(\mathbb{1}(v)|\boldsymbol{x})P_\theta(\boldsymbol{g}_1|v) \\
\tilde{\mathbf{E}}_{u,v}^{(i)} = \mathbf{E}_{u,v} P_\theta(\boldsymbol{g}_{i+1}|v)
\end{cases}
\tag{30}
$$

With passing probability vector $\boldsymbol{p}$ and token probability matrix $\mathbf{G}$ introduced in Section 3.3, equations above can be expressed in matrix form:

$$
\begin{cases}
\tilde{\boldsymbol{p}}^T = \boldsymbol{p}^T \odot \mathbf{G}_{1,:} \\
\tilde{\mathbf{E}}^{(i)} = \mathbf{E} \odot \mathbf{G}_{i+1,:}
\end{cases}
\tag{31}
$$

where $\mathbf{G}_{i+1,:}$ should be broadcast to the size of $\mathbf{E}$. Then we can obtain the expression of Equation 29 in matrix form based on the formula of the sum of transition probabilities in Markov chain:

$$
\begin{aligned}
\mathbb{E}_{\boldsymbol{y}'}[C_{\boldsymbol{g}}(\boldsymbol{y}')] &= \|\tilde{\boldsymbol{p}}^T \cdot \prod_{i}^{n-1} \tilde{\mathbf{E}}^{(i)}\|_1 \\
&= \|(\boldsymbol{p}^T \odot \mathbf{G}_{1,:}) \cdot (\mathbf{E} \odot \mathbf{G}_{2,:}) \cdot ... \cdot (\mathbf{E} \odot \mathbf{G}_{n,:})\|_1
\end{aligned}
\tag{32}
$$

Considering $\mathbf{G}_{i+1,:}$ is broadcast across rows, removing parentheses in Equation 32 will not change the results:

$$
\mathbb{E}_{\boldsymbol{y}'}[C_{\boldsymbol{g}}(\boldsymbol{y}')] = \|\boldsymbol{p}^T \odot \mathbf{G}_{1,:} \cdot \mathbf{E} \odot \mathbf{G}_{2,:} \cdot ... \cdot \mathbf{E} \odot \mathbf{G}_{n,:}\|_1
\tag{33}
$$

## D  ANALYSIS OF HANDLING MULTI-MODALITY

**Generation Fluency** We argue that FA-DAT improves NAT performance by better capturing multi-modality with the fuzzy alignment objective. It helps the model avoid generating a mixture of different translations. To demonstrate that, we measure the fluency of outputs from different models. The perplexity (PPL) score is calculated by a pretrained language model (Ng et al., 2019)[6] with context window size 128. Lookahead decoding is applied to DA-Transformer and FA-DAT.

Table 4: Perplexity scores on **raw** WMT14 EN-DE dataset.

|     | Vanilla-NAT | DA-Transformer | FA-DAT | AT | Reference |
|-----|-------------|----------------|--------|-------|-----------|
| **PPL** | 1909.9 | 495.8 | 468.6 | 411.7 | 395.8 |

As shown in Table 4, autoregressive translation (AT) model achieves comparable fluency with gold reference while vanilla-NAT suffers a significant drop due to the disability to handle multi-modality. DA-Transformer improves NAT fluency by a large margin and proposed FA-DAT further reduces the fluency gap between AT and NAT.

**Sequence Length** As longer sentences tend to have more complex grammatical structures, multi-modality usually occurs in their translation distribution. Motivated by this, we also investigate model performance for sequences of different lengths. We split the test set of WMT14 EN-DE into different buckets based on reference length and report the translation quality of each bucket in Table 5.

Table 5: BLEU scores of different length buckets on **raw** WMT14 EN-DE dataset.

|  | DA-Transformer | FA-DAT |
|--|----------------|--------|
| $L < 20$ | 24.81 | 25.81 |
| $20 \leq L < 40$ | 26.96 | 27.89 |
| $40 \leq L < 60$ | 27.33 | 27.39 |
| $L \geq 60$ | 22.81 | 25.54 |

We find that FA-DAT improves the translation quality of all length buckets, especially for $L \geq 60$, where the gain is 2.73 BLEU. We argue that both word reordering and position shift issues are more common in long sentences due to the syntactic multi-modalities, where the model trained by NLL loss will get confused. As shown in Table 5, NLL-trained DA-Transformer suffers a steep performance drop by around 4 BLEU when translating the bucket of longest sentences. In contrast, FA-DAT shows a much better ability to deal with long sentences, which demonstrates its effectiveness in handling multi-modality.

## E  RESULTS ON DISTILLED DATASET

Though FA-DAT is proposed to deal with the multi-modality problem which is severe on raw training data, we also conduct experiments to evaluate the performance of FA-DAT on distilled data, where the data distribution is much simplified. Following Gu et al. (2018), we use Transformer-*base* as the teacher model to generate the distilled dataset. We apply Lookahead decoding to both DA-Transformer and FA-DAT and report the results in Table 6.

We find that FA-DAT consistently outperforms DA-Transformer on both settings of training data. However, the improvement on the distilled dataset is relatively marginal. We attribute it to that multi-modality in distilled data is already alleviated due to the simplification by the autoregressive teacher. Interestingly, we note that FA-DAT performs better on raw training data by a margin of 0.36 BLEU, which shows a distinct feature in contrast to NAT models in the literature. We argue that traditional sequence-level KD limits the performance of NAT model by imposing an upper

---

[6]https://github.com/facebookresearch/fairseq/tree/main/examples/language_model

Table 6: BLEU scores on **raw** and **distilled** WMT14 EN-DE dataset.

|            | DA-Transformer | FA-DAT |
|------------|:--------------:|:------:|
| **Raw**      | 26.56          | 27.53  |
| **Distilled** | 26.94          | 27.17  |

bound (AT's performance). It demonstrates that NAT model endowed with the ability to handle data multi-modality can benefit more from training on authentic data, showing the potential of further developing stronger NAT models.

## F    MEASURE TRANSLATION QUALITY WITH LEARNED METRICS

Considering fuzzy alignment objective is a natural fit with $n$-gram matching-based evaluation metrics, we additionally measure the translation quality with two learned metrics: BERTScore (Zhang et al., 2020)[7] and BLEURT (Sellam et al., 2020), which have been demonstrated to correlate well with human judgments. For BLEURT, we use the currently recommended checkpoint BLEURT-20[8] (Pu et al., 2021) to generate scores. The results are reported in Table 7 and 8.

Table 7: BERTScores on **raw** WMT14 EN↔DE and WMT17 ZH↔EN dataset.

|                | | EN-DE | DE-EN | ZH-EN | EN-ZH |
|----------------|----------------|:-----:|:-----:|:-----:|:-----:|
| DA-Transformer | +*Greedy*      | 60.86 | 66.91 | 56.70 | 61.64 |
|                | +*Lookahead*   | 60.89 | 67.00 | 56.97 | 61.62 |
|                | +*Joint-Viterbi* | 60.94 | 67.12 | 56.63 | 60.75 |
| FA-DAT         | +*Greedy*      | 62.16 | 67.48 | 58.50 | 62.80 |
|                | +*Lookahead*   | 62.18 | 67.50 | 58.51 | 62.66 |
|                | +*Joint-Viterbi* | 62.17 | 67.49 | 58.12 | 62.53 |

Table 8: BLEURT scores on **raw** WMT14 EN↔DE and WMT17 ZH↔EN dataset.

|                | | EN-DE | DE-EN | ZH-EN | EN-ZH |
|----------------|----------------|:-----:|:-----:|:-----:|:-----:|
| DA-Transformer | +*Greedy*      | 61.63 | 68.06 | 62.14 | 59.52 |
|                | +*Lookahead*   | 61.13 | 68.02 | 61.87 | 58.95 |
|                | +*Joint-Viterbi* | 61.34 | 68.13 | 61.39 | 58.62 |
| FA-DAT         | +*Greedy*      | 64.26 | 68.14 | 62.72 | 60.10 |
|                | +*Lookahead*   | 64.29 | 68.14 | 62.70 | 60.00 |
|                | +*Joint-Viterbi* | 64.26 | 68.32 | 62.44 | 59.94 |

As shown in Table 7 and 8, FA-DAT improves both BERTScore and BLEURT scores on all four translation tasks, which is consistent with the results of BLEU in Table 2. Interestingly, we find Joint-Viterbi decoding does not show its superiority compared to other decoding approaches, as opposed to the results measured by BLEU in Table 2. Moreover, BLEURT tends to prefer translations generated by Greedy decoding, especially for the generation of DA-Transformer. We leave the discussion of those findings in future work.

## G    QUALITY-LATENCY TRADE-OFF UNDER BATCH DECODING

Gu & Kong (2021) has pointed out that the speed advantage of non-autoregressive model shrinks when the size of the decoding batch gets larger. As both DA-Transformer and proposed FA-DAT

---

[7]https://github.com/Tiiiger/bert_score
[8]https://github.com/google-research/bleurt

obtain the best translation performance at the cost of a high upsampling ratio ($\lambda = 8$), the extra overhead of computation and memory can make the speedup degradation worse. To have a better understanding of the problem, we compare the speed-quality tradeoffs under different settings of upsampling ratio and decoding batch size. The results are plotted in Figure 3.

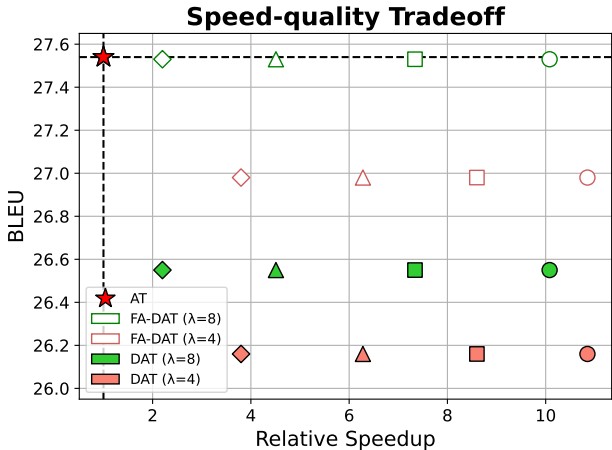

Figure 3: Relative speedup ratio (x-axis) and BLEU score (y-axis) against different settings of batch decoding size (8: ●, 16: ■, 32: ▲, 64: ◆).

It is consistent with previous findings that the speedup drops under a large batch setting. Moreover, the degradation becomes even worse with a larger upsampling ratio. As shown in Figure 3, the relative speedup ratio is reduced to around 2 under decoding batch size 64 with 8 times upsampling model. We attribute this problem to the intrinsic property of the directed acyclic graph, which requires a large number of redundant vertices to model the translation multimodality. However, we find proposed FA-DAT is beneficial to relieve the problem. 4× upsampled FA-DAT achieves a better quality than 8× upsampled DA-Transformer by around 0.5 BLEU with significantly shorter decoding latency. We argue that fuzzy alignment is capable of well-calibrating all the vertices in the graph (as analyzed in Section 4.4), which uses upsampled vertices more efficiently and reduces redundancy.

## H EFFECTS OF PRETRAINING

We further investigate the effects of NLL pretraining in this section.

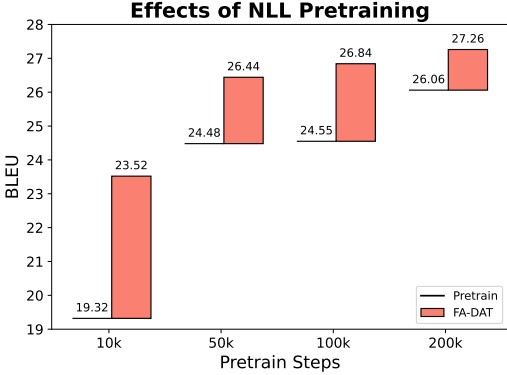

Figure 4: BLEU scores of FA-DAT and NLL-pretrained models under different settings of pretraining step. No checkpoint averaging trick is applied for both pretrained model and FA-DAT.

As discussed in Section 3.4, we initialize FA-DAT with NLL-trained DA-Transformer. In this experiment, we train FA-DAT from models with different settings of the pretraining step. BLEU scores of FA-DAT and NLL-pretrained models under different settings are plotted in Figure 4.

We find that fuzzy alignment training can boost the performance of NLL-pretrained model under all circumstances. Moreover, initializing FA-DAT with a well-NLL-pretrained model can further improve the translation quality. We argue that $n$-gram-based fuzzy alignment objective models word reordering at the cost of ignoring higher (i.e., $> n$) order dependency. However, such ignored information is modeled when trained with NLL loss due to its verbatim alignment property, which makes NLL pretraining complementary and essential to fuzzy alignment training.

