# OpenReview forum: "Fuzzy Alignments in Directed Acyclic Graph for Non-Autoregressive Machine Translation"
_ICLR.cc/2023/Conference — ICLR 2023 poster_

### Official Review · Reviewer_zAdY · 2022-10-22

**Confidence:** 3
**Correctness:** 3
**Technical Novelty And Significance:** 3
**Empirical Novelty And Significance:** 3
**Recommendation:** 6

**Clarity, Quality, Novelty And Reproducibility:**

It would be better if there could be more illustrations (probably figures) for the descriptions of the proposed loss function and the core algorithm, which might allow easier reading.

It seems slightly strange to me that in Equation (9)/(10), suddenly \theta appears as the first argument on the left-hand side.


**Strength And Weaknesses:**

Strength:

The work provides an effective way to calculate expected n-gram counts for the DAG-based decoder.

The results seem impressive, with clear improvements over the baselines and the final FA-DAT models almost close the gaps to auto-regressive ones using raw data.

Weakness:

One concern is that the proposed model needs to use a relatively large lambda (decoder length) to obtain good enough results. This might not be a problem when performing single-instance decoding, while this brings extra operations and might more obviously affect batched decoding.

The proposed method seems a natural fit to evaluating using BLEU, but it is unclear with other MT evaluation metrics. (I’m not sure whether BERTScore is a good metric for MT evaluation).

Some more analyses and ablations might be needed, for example, on the effects of the initialization from NLL-trained DA-Transformer, the adaptation of the brief penalty.


**Summary Of The Paper:**

This work improves the DA-Transformer with an N-gram based fuzzy alignment loss for non-autoregressive machine translation. Instead of enforcing strict monotonic alignment and NLL loss in the original DA-Transformer, the proposed method adopts a fuzzy loss similar to N-gram precision in the calculation of BLEU. With evaluations on standard MT datasets, the proposed method is shown to obtain impressive results using raw (non-distilled) training data, bringing improvements over the DA-Transformer and almost closing the gaps to auto-regressive models.

**Summary Of The Review:**

Overall I think this paper did a relatively good job on providing an effective algorithm for calculating expected n-grams and pushing the raw-training results close to the auto-regressive ones. My main concern is the evaluation of decoding efficiency, where the large decoder length might have an undesired effect when using batched decoding.

---

> ### Author Response · Authors · 2022-11-17
> **Response to Reviewer zAdY**
>
> **We sincerely thank the reviewer for the constructive and helpful feedback. We hope the following responses would help address your concerns.**
>
> **1. Concerns on the high upsampling ratio which may hurt speedup under batch decoding**
>
>    As you pointed out, both DA-Transformer and the proposed FA-DAT require a high upsampling ratio ($\lambda = 8$) to achieve the best translation performance.
>    Such a setting will make the speedup degradation worse when the size of the decoding batch gets larger.
>    We conduct an analysis of this issue in Appendix G.
>    The relative speedup ratio is reduced to around 2 under decoding batch size 64 while the decoder length is 8× upsampled in both DA-Transformer and FA-DAT.
>    We attribute this problem to that DAG requires a large number of redundant vertices to model the multi-modality.
>    However, we find proposed FA-DAT is beneficial to relieve the problem.
>    4× upsampled FA-DAT achieves a better quality than 8× upsampled DA-Transformer by around 0.5 BLEU while the latency under batch decoding is significantly shorter.
>    We argue that fuzzy alignment is capable of well-calibrating all the vertices in the graph, which uses upsampled vertices more efficiently and reduces redundancy.
>
> **2. Concerns on the evaluation metric**
>
>    Following your advice, we also measure the translation quality with BLEURT [1] in Appendix F.
>    In contrast to BLEU, BLEURT is a BERT-based evaluation metric, which is not based on n-gram overlapping and has been demonstrated to correlate well with human judgments.
>    We find that FA-DAT improves BLEURT scores on all four translation tasks, which is consistent with the results of BLEU and BERTScore.
>
>    [1] Sellam et al. BLEURT: Learning Robust Metrics for Text Generation, ACL 2020.
>
> **3. Concerns on the effects of initialization**
>
>    Thanks for your suggestion. We have appended the discussion of the effects of NLL pretraining in Appendix H.
>    We find fuzzy alignment training can boost the performance of NLL-pretrained model under all settings of pretraining.
>    Moreover, initializing FA-DAT with a well-NLL-pretrained model can further improve the translation quality.
>    We argue that n-gram-based fuzzy alignment objective models word reordering at the cost of ignoring higher (i.e., >n) order dependency.
>    However, such ignored information can be modeled when training with NLL loss due to its verbatim alignment property, making NLL pretraining complementary and essential to fuzzy alignment training.
>
> **4. Unclear Eq. 9&10**
>
>    Sorry for the confusion. We have fixed the notations in Eq. 9&10 to make them more straightforward.
>    We use $p_n(\theta,a,y)$ to denote the averaged n-gram precision of model $\theta$'s outputs against reference $y$ when a certain path $a$ is given.
>    We use $p_n(\theta,y)$ to denote the averaged n-gram precision of model $\theta$'s outputs against reference $y$.

---

### Official Review · Reviewer_Qie9 · 2022-10-24

**Confidence:** 4
**Correctness:** 4
**Technical Novelty And Significance:** 3
**Empirical Novelty And Significance:** 3
**Recommendation:** 5

**Clarity, Quality, Novelty And Reproducibility:**

The quality of this work is good.
The clarity of the proposed method still has room for improvement.
The originality of this work is ordinary.

**Strength And Weaknesses:**

strengths:
1. The final performance of FA-DAT is exciting.
2. It is interesting to train NAT models with fuzzy alignment of n-grams, which relaxes the order constraints of n-grams.

weaknesses:
1. The new training method for DAT is too similar to the proposed training method in [1].

[1] Chenze Shao, Yang Feng. Non-Monotonic Latent Alignments for CTC-Based Non-Autoregressive Machine Translation, NeurIPS 2022.

**Summary Of The Paper:**

This paper proposes a new training method for DAT (SOTA NAT model). The authors hold the view that all paths in the graph are fuzzily aligned with the reference sentence. Hence, they train the model to maximize a fuzzy alignment score between the graph and reference. The proposed method is interesting, and the improvement of performance for DAT is exciting.

**Summary Of The Review:**

The performance of the proposed method is exciting, and the experimental content is complete, but the proposed method is innovative.

---

> ### Author Response · Authors · 2022-11-17
> **Response to Reviewer Qie9**
>
> **We sincerely thank the reviewer for the constructive and helpful feedback. We hope the following responses would help address your concerns.**
>
> 1.**Firstly, we have to clarify the common points and differences between [1] and this work.**
> Sequence-level training has been proven effective to model translation multi-modality, which often needs the adaptation of Reinforcement Learning due to the nondifferentiable sequence-level metric. Since RL often suffers from unstable gradient and inefficient sampling, how to efficiently introduce the sentence-level metric to the NAT training is studied by both [1] and this work. **However, the technical details and proposed efficient algorithms of these two works, which are the core contributions, are quite different.**
>
> 2.**Secondly, we would like to point out the fact that [1] was published in the Arxiv later than this work was submitted**, which should be considered as a concurrent work. To alleviate your concerns, we also conduct the experiments of [1] on WMT14 EN-DE raw data with their released codes. The data preprocessing is consistent with that in Section 4.1. Below are the results.
>
> |  Model   |  WMT14 EN-DE |
> |  ----  | ----  |
> | NMLA  [1]| 24.91 |
> | FA-DAT |  27.53 |
>
> We can find FA-DAT handles raw data much better than NMLA. However, we have noticed that NMLA works better in distilled data (Raw 24.91 vs KD 27.57 [1]) while our FA-DAT works better in raw data (Raw 27.53 vs KD 27.17, as discussed in Appendix E). We mark it as a restriction and leave further exploration in the future.
>
> [1] Chenze Shao, Yang Feng. Non-Monotonic Latent Alignments for CTC-Based Non-Autoregressive Machine Translation, NeurIPS 2022.

---

### Official Review · Reviewer_5ZNc · 2022-10-26

**Confidence:** 5
**Correctness:** 3
**Technical Novelty And Significance:** 3
**Empirical Novelty And Significance:** 3
**Recommendation:** 8

**Clarity, Quality, Novelty And Reproducibility:**

The paper is mostly clear, identifies an important problem, and proposes a nice solution.

**Strength And Weaknesses:**

Reasons to accept:
- the intuition is clear and the solution is straightforward
- a novel algorithm with linear complexity is proposed to calculate alignment scores, making it viable to exploit exponential search space.
- the proposed method does not involve knowledge distillation, which is worth noting for NAT. The proposed method keeps the decoding speedup, demonstrating the potential of NAT trained by only raw data.

Reasons to reject:
- it is not clear for how many sentences in the training data suffers from the multi-modality problem. It is not clear whether there are sentences that are more different from the reference, so that the fuzzy alignment does not work at all. It will help if such analysis is provided.

- brief penalty is introduced to prevent the favor for shorter translations. however, it may bring an additional parameter to tune, and make the method less generalize to different datasets or translation distributions. There is no discussion for this issue.

- The experiment section could be improved. This paper proposes the fuzzy alignment for the multi-modality issue, but there are no corresponding experiments for this claim.



Suggestions:

The derivation from eq 6 to eq 7 should be presented clearly (in the main body or in an appendix).

Would authors provide more evidence that your approach does improve the DAT's ability for multi-modality? E.g., can FA-DAT cater for the position shifts and word reorderings issues than baselines?

A few examples may also be helpful for demonstrating the effect of the proposed method.


**Summary Of The Paper:**

To alleviate the problem of multi-modality, the paper introduces a training objective that could consider multiple translations in the directed acyclic graph. The method is based on a fuzzy alignment between reference and the directed acyclic graph based on n-gram matching.  Authors explore an alignment score to measure the fuzzy alignment, where an efficient algorithm in linear complexity is developed to cope with the exponential search space.  Experiments on major WMT benchmarks validate the proposed approach.

**Summary Of The Review:**

I'd like to see this paper accepted, although some more analysis or experiments may be needed.

---

> ### Author Response · Authors · 2022-11-17
> **Response to Reviewer 5ZNc**
>
> **We sincerely thank the reviewer for the constructive and helpful feedback. We hope the following responses would help address your concerns.**
>
> **1. Concerns on brief penalty term which brings additional tunable parameter**
>
>    We are sorry for the confusion. There's no additional hyperparameter or model parameter introduced by brief penalty.
>    Considering FA-DAT is trained to maximize the precision which favors short sentences, brief penalty is essential to eliminating such bias.
>    It only receives reference length and expected output length as inputs, where the latter can be inferred directly from the transition matrix by our proposed algorithm.
>    The formulation of brief penalty is identical in all four translation tasks. We argue that the design would not hurt the generalization ability of our method.
>    We will improve the writing of this part to make it clearer.
>
> **2. Concerns on multi-modality in real training data**
>
>    We did not analyze the multi-modality in the training data due to limited space.
>   However, we notice that prior works have provided analysis of the multi-modality in data both in the view of syntactic structure [1] and of information theory [2], which solidly supports the motivation of this work. We add the discussion of training data multi-modality in the Section of related work.
>
>   [1] Zhang et al. A Study of Syntactic Multi-Modality in Non-Autoregressive Machine Translation, NAACL 2022.
>
>   [2] Huang et al. On the Learning of Non-Autoregressive Transformers, ICML 2022.
>
> **3. Concerns on experiments for the claim of handling multi-modality**
>
>    Thanks for your advice.
>
>    Firstly, we kindly refer reviewers to Table 4, where the statistics provide evidence that FA-DAT properly handles multi-modality in the view of information theory.
>
> NAT's independence assumption limits the expression ability of model distribution, causing $KL(P_{data}(y|x)||P_{NAT}(y|x))>0$ in a multi-modal data regardless of the value of model parameter. Considering such an assumption disables NAT from modeling multi-modal distribution, training NAT with NLL will make the situation even worse. It will achieve perfect approximations on marginal distributions but drops dependency information [1]. **To avoid a mixture of modes being sampled during inference, a theoretically perfect model distribution of NAT should be collapsed to the delta distribution at the point of its best translation.** This motivates us to demonstrate the ability of handling multi-modality by measuring the entropy of NAT's model distribution. Ideally, a well-trained NAT should have a low $H(P_{NAT}(y|x))$.
>
>    **However, estimating $H(P_{NAT}(y|x))$ of a DAG is hard, we turn to compare the entropy values of different models approximately by comparing their $\min \limits_{y} - \log P_{NAT}(y|x)$.**  The reason is that a model with lower entropy should be more confident about its output. As shown in Table 4, $- \log P_{NAT}(y|x)$ of FA-DAT has been reduced significantly, which indicates that the entropy of FA-DAT's distribution is much lower and fuzzy alignment training handles multi-modality properly.
>
>    Following your advice, we have also appended another study in Appendix D to verify the effectiveness of FA-DAT in handling multi-modality. Since longer sentences tend to have more complex grammatical structures, multi-modality usually occurs in their translation distribution.
> Motivated by this, we investigate model performance for different sequence lengths. As shown in Table 6, FA-DAT improves the translation quality of long sentences substantially. We argue that both word reordering and position shift issues are more common in long sentences due to the syntactic multi-modalities, which makes NLL-trained DA-Transformer suffer a steep performance drop by around 4 BLEU when translating the bucket of longest sentences. In contrast, FA-DAT shows a much better ability to deal with long sentences, which further demonstrates its effectiveness in handling multi-modality.
>
>    [1] Huang et al. On the Learning of Non-Autoregressive Transformers, ICML 2022.
>
> **4. Unclear derivation from Eq. 6 to Eq. 7**
>
>    Thanks for your suggestion. We have appended the detailed derivation in Appendix A.

---

### Author Response · Authors · 2022-11-17
**Revised Paper Uploaded**

**We sincerely thank all the reviewers for their time and efforts. Now we have uploaded our revised paper with the following revision:**

1. Added analysis of handling multi-modality in Appendix D

2. Added analysis of latency under batch decoding in Appendix G

3. Added analysis of the effects of NLL pretraining in Appendix H

4. Added results measured by BLEURT in Appendix F

5. Added discussion of data multi-modality in Related Work

6. Added derivation of Eq.7 in Appendix A

7. Fixed notations in Eq.9 &10

---

### Decision · Program_Chairs · 2023-01-20

**Decision:**

Accept: poster

**Justification For Why Not Higher Score:**

I think the contribution, while could be important to translation, is very focused, and could possibly be brittle.

**Justification For Why Not Lower Score:**

The paper stands on solid technical ground.

**Metareview: Summary, Strengths And Weaknesses:**

The reviewers found the paper overall to be a good paper that could contribute to the understanding and state of the art in non-autoregressive translation. There were concerns about the novelty of the work compared to another paper that was recently published in NeurIPS, but I believe the authors responded well to that (especially when considering that NeurIPS happened very recently). The authors were thorough in their responses.

**Note From Pc:**

if the above contains the word "oral" or "spotlight" please see: "oral" presentation means -> notable-top-5% and "spotlight" means -> notable-top-25%. As stated in our emails, we are disassociating presentation type from AC recommendations

**Summary Of Ac-Reviewer Meeting:**

n/a